# Mitocans Revisited: Mitochondrial Targeting as Efficient Anti-Cancer Therapy

**DOI:** 10.3390/ijms21217941

**Published:** 2020-10-26

**Authors:** Lanfeng Dong, Vinod Gopalan, Olivia Holland, Jiri Neuzil

**Affiliations:** 1School of Medical Science, Griffith University, Southport, 4222 Qld, Australia; o.holland@griffith.edu.au; 2School of Medicine, Griffith University, Southport, 4222 Qld, Australia; v.gopalan@griffith.edu.au; 3Institute of Biotechnology, Czech Academy of Sciences, 252 50 Prague-West, Czech Republic

**Keywords:** mitochondrial targeting, anti-cancer strategy, mitocans, drug delivery

## Abstract

Mitochondria are essential cellular organelles, controlling multiple signalling pathways critical for cell survival and cell death. Increasing evidence suggests that mitochondrial metabolism and functions are indispensable in tumorigenesis and cancer progression, rendering mitochondria and mitochondrial functions as plausible targets for anti-cancer therapeutics. In this review, we summarised the major strategies of selective targeting of mitochondria and their functions to combat cancer, including targeting mitochondrial metabolism, the electron transport chain and tricarboxylic acid cycle, mitochondrial redox signalling pathways, and ROS homeostasis. We highlight that delivering anti-cancer drugs into mitochondria exhibits enormous potential for future cancer therapeutic strategies, with a great advantage of potentially overcoming drug resistance. Mitocans, exemplified by mitochondrially targeted vitamin E succinate and tamoxifen (MitoTam), selectively target cancer cell mitochondria and efficiently kill multiple types of cancer cells by disrupting mitochondrial function, with MitoTam currently undergoing a clinical trial.

## 1. Introduction

Mitochondria are dynamic intracellular organelles with their own DNA (mitochondrial DNA, mtDNA). They have multiple important functions, including controlling adenosine triphosphate (ATP) generation, metabolic signalling, proliferation, redox homeostasis, and promotion/suppression of apoptotic signalling pathways. Genetic and/or metabolic alterations in mitochondria contribute to many human diseases, including cancer [1]. Although glycolysis was traditionally considered as the major source of energy in cancer cells, consistent with the so-called “Warburg effect” first suggested almost a century ago, referring to the elevated uptake of glucose that characterizes the majority of cancers, the mitochondrial function known as oxidative phosphorylation (OXPHOS) has been recently recognized to play a key role in oncogenesis [2,3]. In addition, cancer cells uniquely reprogram their cellular activities to support their rapid proliferation and migration, as well as to counteract metabolic and genotoxic stress during cancer progression [4]. Thus, mitochondria can switch their metabolic phenotypes to meet the challenges of high energy demand and macromolecular synthesis [5]. Moreover, cancer cell mitochondria have the ability to flexibly switch between glycolysis and OXPHOS to improve survival [2]. Furthermore, the electron transport chain (ETC) function is pivotal for mitochondrial respiration, and that ETC function is also necessary for dihydroorotate dehydrogenase (DHODH) activity that is essential for de novo pyrimidine synthesis [6]. Recently, the importance of mitochondria in intercellular communication has been further supported by observations that mtDNA within whole mitochondria are mobile and can undergo horizontal transfer between cells. Our group discovered that cancer cells devoid of their mtDNA and therefore lacking their tumorigenic potential could re-gain this property by acquiring healthy mtDNA from host stromal cells via the transfer of whole mitochondria, resulting in a recovery of mitochondrial respiration [7,8] (Figure 1). We also found that respiration is essential for DHODH-dependent conversion of dihydroorotate to orotate, a rate-limiting step of pyrimidine biosynthesis, pointing to an indispensable function of DHODH in tumorigenesis [9].

We have recently proposed the term ‘mitocans’, an acronym derived from the terms mitochondria and cancer, a group of compounds with anti-cancer activity exerted via their molecular targets within mitochondria, some mitocans being selective for malignant tissues [10]. This classification has been used by others, as exemplified by a recent paper [11]. These various agents targeting mitochondria and their various functions contribute to novel anti-cancer strategies with high therapeutic potential. These strategies include agents that target ETC and OXPHOS, glycolysis, the tricarboxylic acid (TCA) cycle, apoptotic pathways, reactive oxygen species (ROS) homeostasis, the permeability transition pore complex, mtDNA as well as DHODH-linked pyrimidine synthesis [12,13]. Increasing numbers of studies focus on delivering anti-cancer drugs to mitochondria to treat cancers, and this innovative approach holds great hope for the development of new efficient anti-cancer therapeutics [14,15,16,17].

## 2. Targeting Mitochondrial Metabolism

Mitochondrial metabolism is highly complex and involves multiple functions and signalling pathways. The major functions of mitochondria are the production of ATP via OXPHOS and formation of metabolites needed to meet the bioenergetic and biosynthetic demands of the cell. Mitochondria are also central to a wide variety of vital cellular processes including apoptosis, maintenance of calcium homeostasis, redox signalling, steroid synthesis, and lipid metabolism. In addition, mitochondria have the ability to alter their bioenergetic and biosynthetic functions to meet the metabolic demands of a cell via a cross-talk with other sub-cellular organelles, in particular the nucleus, but also the endoplasmatic reticulum [18]. Accumulating evidence suggests that mitochondrial functions, including bioenergetics, biosynthesis, and signalling are essential for tumorigenesis. Therefore, targeting mitochondrial metabolism presents a broad spectrum of strategies to fight cancer. There are two basic modes of communication between mitochondria and the rest of the cell: anterograde and retrograde signalling. Anterograde signalling denotes signal transduction from the cytosol (and the various components contained by it) to mitochondria, retrograde signalling refers to signal transduction from mitochondria to the cytosol. There are multiple mechanisms of retrograde signalling, including the release of metabolites and ROS. In many types of cancer cells, large amounts of ROS are produced by mitochondrial ETC through oxidative metabolism. Mitochondrial ROS then activate signalling pathways proximal to mitochondria to promote cancer cell proliferation and tumorigenesis [19].

### 2.1. Targeting Mitochondrial Electron Transport Chain Function

Mitochondrial ETC comprises four complexes (I–IV) that transfer electrons and engage in redox reactions. Transfer of electrons by means of ETC is coupled to pumping of protons from the matrix to the intermembrane space by complexes I, III, and IV. Functional ETC supports OXPHOS activity and ATP generation that is essential for tumorigenesis (as well as normal cell function). Since the majority of ATP in tumour cells is produced by mitochondria [20,21], targeting mitochondrial ETC function and ATP production might be an effective strategy for cancer therapy. It was reported that drugs blocking mitochondrial ATP production induce cell death in poorly perfused tumours in nutrient-poor environments with limited glucose and oxygen [22]. These tumours show a strong dependence on OXPHOS for ATP generation [12,13]. The natural product papuamine was shown to inhibit ATP production in non-small cell lung cancer (NSCLC) cells to deplete intracellular ATP by causing mitochondrial dysfunction, and to increase mitochondrial superoxide generation with ensuing induction of NSCLC cell apoptosis [23].

Many ETC inhibitors, including metformin, tamoxifen, α-tocopheryl succinate (α-TOS) and 3-bromopyruvate (3BP), act by disrupting the function of respiratory complexes of ETC and by inducing production/accumulation of high levels of ROS to kill cancer cells [24,25]. Metformin, an anti-diabetic drug, has been shown to possess an anti-cancer effect by targeting mitochondrial ATP production without invoking toxicity in normal tissues [18]. The anti-tumorigenic effect of metformin via the inhibition of the mitochondrial respiratory complex I (CI) was also demonstrated [26,27,28]. Tamoxifen is used for the treatment of both early and advanced estrogen receptor-positive (ER+) breast cancer in pre- and post-menopausal women [29]. Recent discoveries suggest that tamoxifen inhibits oxygen consumption via suppression of mitochondrial CI-dependent respiration, which is linked to its anti-cancer activity [30]. Lee and colleagues found that the combination of gossypol and phenformin showed an anti-cancer effect in NSCLC via inhibition of aldehyde dehydrogenase and CI function, and efficiently reduced OXPHOS [31]. Kurelac and colleagues reported that inhibition of mitochondrial CI as an anti-cancer approach yielded promising results in subcutaneous osteosarcoma xenografts [32].

Our group reported mitochondrial complex II (CII) as a novel target for cancer therapy. We showed that α-TOS, an efficient anti-cancer agent, inhibits succinate quinone oxidoreductase (SQR) and succinate dehydrogenase (SDH) activity of mitochondrial CII by interacting with the proximal and distal ubiquinone (UbQ)-binding site [33,34]. Gracillin, a mitochondria-targeted anti-tumour agent, was shown to have broad-spectrum inhibitory effects on the viability of a large panel of human cancer cell lines by disruption of CII function via abrogating SDH activity [35].

Other than inhibition of CI or CII function, some anti-cancer compounds affect mitochondrial complex IV (CIV) or ATPase (CV) activity, inhibiting cancer cell respiration and ATP production. A small molecule VLX600 was reported to be active against colon cancer by disrupting the function of CIV, by suppressing the expression of its subunit 1, the COX-1 [36]. Tigecycline, an FDA-approved agent targeting leukemic cells, significantly decreased the activity of CI and CIV of cancer cells and selectively killed leukaemia stem and progenitor cells, while sparing normal hematopoietic cells [37]. Gamitrinib, a small-molecule inhibitor of ATPase selectively accumulated in mitochondria, diminished mitochondrial ATP production and displayed anti-tumorigenic properties in experimental models of cancer [38]. Mitotane has been used for the treatment of adrenocortical cancer and elicits its anti-cancer effects via inhibition of mitochondrial respiration. It was also used to target mitochondria and to induce apoptosis in thyroid cancer treatment [39].

### 2.2. Targeting Tricarboxylic Acid (TCA) Cycle

The TCA cycle, also known as the Krebs cycle, is located in the mitochondrial matrix in eukaryotic cells. It comprises a series of chemical reactions used by aerobic organisms to release stored energy via oxidation of acetyl-CoA derived from carbohydrates, fats, and proteins. The TCA cycle is a source of electrons that feed into ETC to drive the electrochemical proton gradient required for ATP generation. Its intermediates are used for biosynthesis of various macromolecules. This is exemplified by glutamine, a major carbon source that replenishes the TCA cycle intermediates and sustains their utilization for biosynthesis in tumour cells [40]. It is converted into glutamate, which is further converted into α-ketoglutarate that is required in a range of processes, including generation of the TCA cycle reducing equivalents NADH and FADH2, which are used by ETC to generate ATP [16]. Many cancer cells exhibit addiction to glutamine, such that targeting glutamine catabolism could be a plausible anti-cancer strategy. Specific glutaminase inhibitors, such as tetrahydrobenzo derivative-968 and BPTES, inhibit glutamine catabolism and delay tumour growth in experimental cancer models [41,42]. Inhibiting the conversion of glutamate to α-ketoglutarate can also suppress tumour growth [43,44]. Isocitrate dehydrogenase 1 and 2 (IDH1, IDH2) catalyze the conversion of isocitrate to α-ketoglutarate, playing a critical role in tumorigenesis [11]. IDH1 and IDH2 have been found mutated in multiple human cancers [18], rendering them as promising targets for anti-cancer therapy. Inhibitors of IDHs including 3BP, dichloroacetate, AGI-5198, and AGI-6780, possess high anti-cancer potential in a broad range of cancer types [10,12,45,46].

### 2.3. Targeting Glycolysis and OXPHOS

Cancer cells efficiently use both glycolysis and OXPHOS for their energy needs. Moreover, malignant cells have the ability of flexibly switching between glycolysis and OXPHOS, and this feature plays a major role in multiple modes of resistance to oncogenic inhibition [3,12]. Agents that target both glycolysis and OXPHOS may be considered as potentially efficient anti-cancer therapeutics. Combining glycolytic inhibitors together with mitochondria-targeted agents synergistically suppresses tumour cell proliferation [13].

Hexokinase II (HKII) is a major isoform of enzyme overexpressed in cancer cells with an important role in maintaining glycolytic activity. It also associates with the voltage-dependent anion channel (VDAC) on the mitochondrial outer membrane that has a function in apoptosis. As such, the inhibition of HKII will not only inhibit glycolysis but may also suppress the anti-apoptotic effect of the HKII–VDAC interaction. FV-429, an inhibitor of hexokinase, strongly induced apoptosis in cancer cells by both inhibition of glycolysis via suppression of HKII and by impairing the mitochondrial function via interfering with the HKII-VDAC interaction, leading to activation of mitochondria-mediated apoptosis [2]. As mentioned previously, metformin is a drug commonly used to treat diabetes, but it also has the ability to suppress multiple types of cancer [47]. It was shown that metformin inhibits HKII in lung carcinoma cells, leading to decreased glucose uptake and its phosphorylation [48]. Combining metformin with 2-deoxyglucose (2-DG), a glycolysis inhibitor, depleted ATP in a synergistic manner and showed a strong synergy for the combined therapeutic effect in pancreatic cancer cells [13]. Mitochondria-targeted carboxy-proxyl (Mito-CP) in combination with 2-DG led to significant tumour regression, suggesting that the dual targeting of mitochondrial bioenergetic metabolism and glycolysis may offer a promising chemotherapeutic anti-cancer strategy [3]. When combined with 2-DG, the anti-cancer effect of the BH3 mimetic ABT737 was significantly potentiated in human ovarian cancer cells [49]. Inducing mitochondrial uncoupling, where mitochondrial membrane potential is dissociation from ATP formation, is a new strategy with potential anti-cancer activity, as it promotes pyruvate influx to mitochondria and reduces various anabolic pathway activities. Indeed, the induction of mitochondrial uncoupling inhibits cell proliferation and reduces the clonogenicity of cultured colon cancer cells [50].

In recent years, there has been an upsurge in research focusing on reprogramming cancer cells via the understanding of their metabolic ‘signatures’. Alterations in mitochondrial bioenergetics and impaired mitochondrial function may serve as effective targeting strategies such as in triple-negative breast cancer (TNBC), where hormone receptors are absent and endocrine therapy inefficient. Glucose starvation of MDA-MB-231 and MCF-7 breast cancer cells provoked a decreased mitochondrial respiration. Glucose starvation also sensitized MDA-MB-231 cells to apoptosis and decreased their migratory potential [51].

## 3. Targeting Mitochondrial Redox Signalling Pathways and ROS Homeostasis

Tumour cells can alter their redox balance and deregulate redox signalling to support malignant progression and to gain resistance to treatment [51]. They increase their antioxidant capacity to counterbalance the increased production of ROS [52,53], which permits them to generate high levels of ROS to activate proximal signalling pathways that promote proliferation and would not otherwise induce cancer cell death or senescence. Mitochondria produce high levels of ROS that are functional for multiple signalling networks underlying tumour proliferation, survival, and metastatic process [54]. Disturbing redox signalling pathways and breaking up ROS homeostasis in cancer cells could be used in cancer therapy. Thus, strategies aimed at altering redox signalling events in tumour cells and intended to disable key antioxidant systems in the presence of ROS inducers may represent promising new anti-cancer treatments [55].

### 3.1. Targeting Redox-regulating Enzymes and ROS Production

ROS are short-lived molecules with unpaired electrons derived from partially reduced molecular oxygen that are constantly generated, transformed, and eliminated via a variety of cellular processes including metabolism, proliferation, differentiation, immune system regulation, and vascular remodelling [56]. The level of ROS is critical for cell survival and cell death. At moderate concentrations, ROS activates the cancer cell survival signalling, while a high level of ROS can cause damage and induce apoptosis in cells. ETC is the major site of ROS production, and high levels of ROS released due to interference with ECT complexes cause cellular damage. Promoting mitochondrial ROS production to induce cancer cell death could thus enhance the efficacy of chemotherapy [47,55]. Oxymatrine was reported to efficiently kill human melanoma cells by generating high levels of ROS [12]. Capsaicin, casticin, and myricetin display anti-cancer activity by increasing ROS generation, leading to the disruption of mitochondrial transmembrane potential in cancer cells [12]. A novel mitochondria-targeted fluorescent probe BODIPY-TPA (in which triphenylamine, TPA, is coupled to the fluorophore) was shown to induce apoptosis in gastric cancer via disruption of the mitochondrial redox balance and ROS accumulation [15].

Of high biological relevance, nicotinamide adenine dinucleotide phosphate (NADPH) oxidases are enzymes that catalyze the production of O_2_^−^·or H_2_O_2_ using NADPH as a reductant [57]. ETC uses NADH and FADH_2_ to generate O_2_^−^ by means of univalent reduction of molecular oxygen resulting in electron leakage during mitochondrial respiration [58,59,60]. NADPH generation occurs in mitochondria from one-carbon metabolism [61] that is initiated by serine hydroxymethyltransferase 2 (SHMT2). As a key enzyme in serine/glycine biosynthesis and one-carbon metabolism, SHMT2 was shown to play a role in tumour growth and progression in many cancer types [62]. Thus, lowering SHTM2 levels decreased tumour growth [63]. Another enzyme involved in mitochondrial one-carbon metabolism, methylenetetrahydrofolate dehydrogenase/cyclohydrolase (MTHFD2), may represent a viable therapeutic target in cancer, since the loss of MTHFD2 increases ROS levels and sensitizes cancer cells to oxidant-induced cell death [64]. Furthermore, targeting mitochondrial one-carbon metabolism enzymes together with other therapies known to increase ROS may have potential benefit in cancer treatment [65]. In addition, Wang and colleagues synthesized binuclear Re(I) tricarbonyl complexes ReN and ReS that accumulate in mitochondria and cause oxidative stress and mitochondrial dysfunction, which have been shown to slow down the bioenergetic rate to inhibit tumour growth [66].

### 3.2. Targeting Mitochondrial Apoptotic Signalling Pathways

The intrinsic apoptotic signalling pathway refers primarily to mitochondria-mediated apoptotic pathways, in which Bcl-2 family proteins (e.g., Bcl-2, Bcl-x_L_ and Bax) play pivotal roles. The intrinsic apoptotic signalling pathway is mediated by insertion of pro-apoptotic proteins Bax/Bak into the outer mitochondrial membrane. Subsequently, cytochrome c is released from the mitochondrial intermembrane space into the cytosol [67]. Cytochrome c combines with Apaf-1 and procaspase-9 to form the apoptosome, which triggers caspase-9 activation followed by the activation of caspase-3, which leads to cell death [68,69]. Bcl-2 and Bcl-x_L_ are anti-apoptotic proteins which prevent the release of cytochrome c and protect cells from apoptosis [70]. Targeting Bcl-2 family proteins can be therefore used as anti-cancer strategy via activation of the apoptotic signalling pathway in cancer cells. Navitoclax, TW-37, GX15-070, and BM-1197 are Bcl-2 or Bcl-x_L_ inhibitors with anti-cancer activity in a broad range of cancer types [12]. Venetoclax, another Bcl-x_L_ inhibitor (a BH3 mimetic), has been approved for use in patients with lymphoma and chronic lymphocytic leukaemia [16,71]. Other compounds such as Gossypol, Navitoclax, ABT-737 and α-TOS act as mimetics of the Bcl-2 homology-3 domain to kill cancer cells via activation of post-mitochondrial apoptotic signalling [10]. Matrine was used to treat acute lymphoblastic leukaemia by ROS generation, and the agent significantly up-regulates the pro-apoptotic protein Bax and down-regulates the anti-apoptotic Bcl-2 protein [72]. ECPU-0001, an efficient tumoricidal agent, exhibited impressive anti-cancer activity and translated to the treatment of lung adenosarcoma by targeting the Bcl-2-associated intrinsic pathway of apoptosis [73]. SWNH treatment was reported to alter the expression of multiple mitochondrial apoptotic pathway-associated proteins and induced apoptosis in hepatoblastoma cells [74]. Silver(I) phosphine acts as an effective chemotherapeutic drug, killing malignant esophageal cells by targeting the mitochondrial intrinsic cell death pathway via lowing levels of ATP, altering ROS activity, and depolarizing the mitochondrial membrane, which leads to a release of cytochrome c and activation of caspase-9 [75].

Since Akt/PKB can inactivate pro-apoptotic factors such as Bad and procaspase-9 [76], activation of the kinase has been related to increased resistance of prostate cancer cells to apoptosis [77]. Akt/PKB activates the IκB kinase (IKK), which is a positive regulator of the survival transcription factor NFκB, and it has been shown that Akt/PKB links NFκB to modulation of anti-apoptotic effects in lymphoma cells [78]. Recent research has focused on targeting the m-TOR/PI3K/Akt signalling pathway to induce cancer cell apoptosis. Zhu et al. found that Galangin increased expression of Bax and cytochrome c and decreased expression of Bcl-2, resulting in the demise of renal cancer cells. It may also inhibit migration and invasion of kidney cancer cells and suppress the expression of several important proteins of the PI3K/Akt/m-TOR signalling pathway [79]. Pterostilbene exerted potent anti-tumour effects in HeLa cervical cancer cells by disrupting mitochondrial membrane potential, apoptosis induction, and targeting the m-TOR/PI3K/Akt pathway [80]. In addition, icariin was shown to inhibit the growth of human cervical cancer cells by inducing apoptosis and autophagy via the m-TOR/PI3K/Akt pathway [81].

## 4. Targeting Other Signalling Pathways that Affect Mitochondrial Functions

### 4.1. p53 Signalling Pathway

The tumour suppressor protein p53 has emerged as a key regulator of metabolic processes and metabolic reprogramming in cancer cells. p53 engages in the mitochondrial cell death machinery and plays an important role in cell survival and function [82]. p53 has been shown to modulate mitochondria-linked programmed cell death [83,84]. One of its ‘targets’ is the pro-apoptotic protein Bax, whose expression is controlled by p53 [85]. Proline dehydrogenase, a p53-inducible inner-mitochondrial membrane flavoprotein linked to electron transport for anaplerotic glutamate and ATP production, is a unique mitochondrial cancer target. N-PPG-like inhibitors of proline dehydrogenase could suppress multiple types of breast cancer cell growth [86]. Qin et al. reported that tacrine platinum(II) complexes exhibited cytotoxic activity in NCIeH460, Hep- G2, SK-OV-3, SK-OV-3/DDP and MGC80-3 cancer cells and induced cell apoptosis by means of activation of the p53 signalling pathway and dysfunction of mitochondria [87].

### 4.2. EGFR-Targeting via Mitochondria-Mediated Apoptosis

The novel recombinant EGFR-targeting β-defensin Ec-LDP-hBD1 displays both selectivity and enhanced cytotoxicity against cancer cells by inducing mitochondria-mediated apoptosis and exhibiting high therapeutic efficacy against EGFR-expressing carcinoma xenografts. This novel format of β-defensin, which induces mitochondrial-mediated apoptosis, is likely to play an active role in EGFR-targeting cancer therapy [88].

### 4.3. Mitochondrial Fission

Mitochondria are dynamic organelles frequently undergoing fission and fusion cycles to maintain their integrity. Disruption of mitochondrial dynamics plays a role in cancer progression. Therefore, proteins involved in regulating the homeostasis of fission and fusion are potential targets for cancer treatment. mDIVI1 is an inhibitor of the mitochondrial fission protein DRP1, which can be used for elimination of cancer stem cells [89]. IR-783, a near-infrared heptamethine cyanine dye, has been reported to exert anti-cancer effects; linked to this, IR-783 was shown to cause induction of mitochondrial fission in MDA-MB-231 and MCF-7 cells, and to lower the levels of ATP [90].

### 4.4. Targeting Mitochondrial DNA (mtDNA)

Cancer is characterised by altered energy metabolism involving not only genetic alternations in nDNA but also mtDNA mutations and changes in mtDNA copy number [91,92,93,94]. It has been shown that somatic mtDNA alterations or low mtDNA copy number promote cancer progression and metastasis via activation of mitochondrial retrograde signalling [95,96]. Eliminating mtDNA limits tumorigenesis [97]; emerging studies from our group have shown that mtDNA plays an essential role in cancer progression, such that mtDNA-depleted cancer cells fail to form tumours, and these cells have to acquire mtDNA from the host by means of horizontal transfer of whole mitochondria to regain their tumorigenic ability [7,8,98] (Figure 1). Importantly, mitochondrial transfer has also been found to occur following mitochondrial damage by chemotherapy and radiation treatment to better protect cancer cells from aberrant physiology [99,100,101]. Therefore, targeting mtDNA and/or blocking mitochondrial transfer presents a novel strategy that may overcome drug resistance and enhance cancer therapy.

It has been reported that cyclomethylated Ir(III) complexes can intercalate into mtDNA and induce mtDNA damage, followed by a decline of mitochondrial membrane potential, suppression of ATP generation, and disruption of mitochondrial energetics and metabolic status, eventually causing cancer cell apoptosis [102]. Additionally, using an mtDNA-depletion model, we found that DHODH-driven pyrimidine biosynthesis is an essential pathway which links respiration to tumorigenesis, demonstrating that DHODH could be a potential wide-spectrum target for cancer therapy [9].

## 5. Mitochondria-Specific Anti-Cancer Drug Delivery

As mentioned, mitochondria are plausible targets for anti-cancer strategies. Agents that target mitochondrial metabolism, the ETC, apoptotic pathways as well as other mitochondrial-linked signalling pathways, show efficient anti-cancer potential. Many anti-cancer drugs (doxorubicin, cisplatin, paclitaxel, resveratrol) are already known to act within the membrane and the matrix of mitochondria [103,104]. Delivering drugs directly to mitochondria greatly enhances their anti-cancer efficacy [105]. Therefore, mitochondria-oriented delivery of anti-cancer drugs has become a focus of recent research, with the expectation to improve anti-cancer efficiency of chemotherapeutics and to overcome drug resistance. Currently, there are two well-known approaches for mitochondrial drug delivery: direct conjugation of the targeting ligand/moiety to drugs and attachment of the targeting ligand to a nanocarrier [106].

### 5.1. Direct Conjugation of Mitochondria-Targeting Ligands to Drugs

A number of direct conjugates have been reported for mitochondrial delivery of anti-cancer drugs using various targeting moieties, including lipophilic cations (triphenylphosphonium; rhodamine 123; and dequalinium) and peptides (mitochondria-penetrating peptide (MPP), mitochondria-targeting sequence (MTS) peptide, and Szeto-Schiller (SS) peptides) [106]. In this paragraph, we will focus on the lipophilic cations targeting moieties.

There are multiple mechanisms and techniques to deliver drugs into mitochondria using the well-known approach based on a higher mitochondrial membrane potential of cancer cells compared to that of their cytosol and non-cancer cells, which allows selective targeting of cancer cell mitochondria [107]. Triphenylphosphonium (TPP), as a delocalized lipophilic cation, is a frequently used mitochondria-targeting molecule, and a number of studies have used this ligand to develop mitochondria-targeted anti-cancer drugs (Figure 2). Our group developed and tested TPP-conjugated drugs due to TPP’s strong mitochondrial targeting ability; these agents belong amongst ‘mitocans’ that we defined earlier [10]. More specifically, TPP-tagged mitocans are agents which selectively accumulate in mitochondria of cancer cells, mostly causing ROS generation with ensuing apoptotic cell death (Figure 3) [10]. Within this class of compounds, we have synthetized to date mitochondria-targeted vitamin-E succinate (MitoVES), mitochondria-targeted tamoxifen (MitoTam) and mitochondria-targeted metformin (MitoMet), all of which show superior anti-cancer activity compared to the parental compounds. MitoVES disturbs the function of mitochondrial CII, MitoTam and MitoMet target mitochondrial CI, and in all the cases this results in the formation of high levels of ROS that leads to cancer cells death [15,108,109,110,111,112]. Of note, MitoTam has been tested in a Phase 1 trial (EudraCT 2017-004441-25) with promising outcomes, and we are currently extending this into a Phase 2 trial.

A number of researchers have used TPP+ conjugation to deliver anti-cancer drugs to mitochondria. Bryant and colleagues reported that Hsp90-TPP showed a 17-fold increase in mitochondrial accumulation than Hsp90 itself, and that “mitochondrial Hsp90” efficiently killed both primary and cultured acute myeloid leukaemia cells [113]. Han and colleagues synthesized TPP-doxorubicin (TPP-Dox) and found that it was taken up at a higher rate than free Dox by MDA-MB-435 Dox-resistant cells, indicating that TPP-Dox conjugate was able to overcome drug resistance [114]. Two phenol TPP-derivatives were shown to have remarkable cytotoxic activity against different cancer cell lines, with lower toxicity against normal cells [115]. Chlorambucil is an anti-cancer agent that damages DNA. Millard and colleagues synthesized a TPP-chlorambucil conjugate and found that it accumulated in mitochondria, leading to mtDNA damage and significant suppression of tumour progression. TPP-chlorambucil showed about an 80-fold enhancement of cancer cell-killing activity in a panel of breast and pancreatic cancer cell lines that are largely insensitive to the parent drug [116]. A dual fluorescent mitochondrial targeting F16–TPP analogues also showed a promising therapeutic effect in cancer cells [117]. A modification of a pro-apoptotic peptide with two mitochondria-targeting TPP moieties caused its efficient accumulation in mitochondria of cancer cells, inducing mitochondrial dysfunction and triggering mitochondria-dependent apoptosis to efficiently eliminate cancer cells [118]. Wang and Xu reported that TPP-coumarin as a novel mitochondria-targeted drug effectively inhibited HeLa cell proliferation and triggered apoptosis by promoting ROS generation and mitochondrial Ca2+ accumulation [119].

Recently, photodynamic therapy (PDT) has been proven to be a minimally invasive and highly efficient therapeutic strategy of cancer treatment. TPP was used in the development of a group of photosensitizers to enhance their cancer cell uptake efficacy and mitochondrial localization. Noh and colleagues developed MitDt, a mitochondrial targeting photodynamic therapeutic agent, by conjugating the heptamethine mesoposition of a cyanide dye with TPP. The PDT effects of MitDt are amplified after laser irradiation because mitochondria are susceptible to ROS which triggers anti-cancer effects [17]. The cationic TPP-octahedral molybdenum cluster complex was shown to rapidly internalize into HeLa cells and accumulate in their mitochondria, triggering intensive phototoxic effect from the 460 nm irradiation [120]. Lei and colleagues reported that TPP-porphyrin photosensitizers with photodynamic activity present significant phototoxicity at concentrations at which “dark toxicity” is negligible towards human breast cancer cells [121]. Also, the AgBiS2-TPP nanocomposite was reported as applicable in photothermal therapy, and it was demonstrated as an agent with high anti-cancer activity [122]. Besides TPP-conjugated drugs, rhodamine derivatives and guanidine-drug conjugates also present as mitochondria-targeting anti-cancer agents that accumulate in mitochondria based on their lipophilic and cationic properties [123,124,125,126,127].

### 5.2. Mitochondria-Targeting Ligands and Nanocarrier (Mitochondria-targeted Nanocarriers)

Nanocarriers have been considered to carry drugs and deliver them to the target areas of tissues to enhance drug efficiency and reduce toxicity. Nanocarriers include micelles, polymers, carbon-based materials, liposomes, metallic nanoparticles, and dendrimers that all have been developed for applications, particularly in the field of chemotherapeutic drug delivery [106,128]. The size of the nanocarriers should be small, ideally within the range of 10–200 nm in diameter, so they can deliver drugs to otherwise inaccessible sites within various tissues. For mitochondrial targeting, a nanoparticle needs to be tagged with a targeting ligand, which preferentially delivers drugs to mitochondria. However, in some cases, the nanoparticle itself can act as a mitochondria-targeting agent, based on its properties. Similar to the concept of direct targeting, cationic ligands such as dequalinium (DQA) and TPP are often attached to nanocarriers to generate mitochondria-targeted nanocarriers (MTNs). These MTNs can overcome solubility, selectivity, and resistance issues of individual drugs, and accumulate primarily in mitochondria to improve their therapeutic effect. The first nanomaterial applied for mitochondrial targeting was presented by DQA micelles (DQAsomes), which exhibit liposome-like self-assembly properties in aqueous solutions [129,130]. They have higher cell killing activity in cancer cells compared to normal cells, resulting from selectively enhanced ROS generation, disruption of mitochondrial transmembrane potential, and blockade of ATP synthesis [131,132].

Doxorubicin (Dox) is one of the first choices of chemotherapeutic drugs applied to the nanocarrier delivery system. Liu and colleagues prepared Dox-loaded TPP-lonidamine self-assembled nanoparticles (NPs), which contain polyethylene glycol groups to enhance their circulation in blood for more extended periods. The NPs showed greater cytotoxicity in both drug-sensitive and drug-resistant cancer cells compared to Dox [133]. Using a hydrazone bond, a hyaluronic acid-Dox-TPP conjugate was prepared to specifically deliver TPP-Dox to mitochondria. A cell uptake study showed more significant mitochondrial accumulation of the NPs in MCF/ADR (Adenocarcinoma) cells, and further cytotoxicity and anti-tumor studies confirmed their enhanced efficacy compared to free Dox and TPP-Dox conjugates [134].

Zhang et al. used glycyrrhetinic acid-attached graphene oxide with Dox as a model drug for dual targeting to mitochondria and the cell membrane due to its ability to interact with the mitochondrial respiratory chain and high binding affinity to protein kinase C (PKC) α, which is overexpressed in certain cancer types [135,136,137]. Carbon quantum dots (CQDs) have been used as fluorescent probes for bioimaging/biolabeling and biosensing due to their stable and robust fluorescence and low toxicity [138]. Mitochondria-targeting Dox-loaded CQD nanoparticles are expected to overcome drug resistance. D-α-Tocopherol polyethylene glycol succinate (TPGS), an inhibitor of the permeability glycoprotein (Pgp, a multidrug resistance protein), was included in the NP to inhibit Pgp expression in drug-resistant cancer cells. TPP was conjugated to TPGS and covered the CQDs. The cytotoxicity results revealed that Dox-loaded CQD NPs had a five-times lower IC50 value in drug-resistant MCF7 cells compared to free Dox [139]. Furthermore, it has been reported that DQA-Dox-containing micelle NPs have up to 5-fold greater tumour suppression effects than free Dox in a Dox-resistant tumour model [140].

Lee and colleagues reported the formation of aggregates off a TPP-tagged coumarin probe (TPP-C) in an aqueous solution. With the encapsulation of Dox into the TPP-C NPs, the anti-cancer drug was efficiently delivered to the mitochondria and exerted considerable cytotoxicity toward cancer cells [141]. Lonidamine (LND) can act on mitochondria and inhibit energy metabolism in cancer cells and therefore has been used together with chemotherapeutic drugs for synergistically enhanced therapeutic efficacy. However, its use is hindered by poor solubility and slow diffusion in the cytoplasm. Aqueous dispersible NPs containing TPP and LND plus Dox were prepared for synergistic cancer treatment and for overcoming drug resistance. TPP-LND-DOX NPs promote the mitochondrial apoptotic pathway and contribute to the overcoming of drug resistance in cancer therapy [133]. Similarly, TPP-linked lipid-polymer hybrid NPs (DOX-PLGA/CPT) were coated with an acidity-triggered cleavable polyanion (PD) and formed DOX-PLGA/CPT/PD structures. The surface negative charge of DOX-PLGA/CPT/PD prevented their rapid clearance from the circulation and improved their accumulation in tumour tissue via an enhanced effect on permeability and retention. Hydrolysis of amide bonds in PD in weakly acidic tumour tissue leads to the conversion of DOX-PLGA/CPT/PD to positively charged DOX-PLGA/CPT, the latter form eventually accumulated in tumour mitochondria. This results in targeting of mtDNA and induction of tumour cells apoptosis and in overcoming Dox resistance of MCF-7/ADR breast cancer [142].

Based on mesoporous silica nanoparticles (MSNs), a novel enzyme-responsive, multistage-targeted anti-cancer drug delivery system which possessed both CD44-targeting and mitochondrial-targeting properties was developed by Naz and colleagues [143]. First, TPP was attached to the surface of the MSNs, and Dox was then encapsulated into the pore of the MSNs followed by its capping with tumour-targeting molecules of hyaluronic acid (HA). The final product consists of Dox-loaded, TPP-attached, HA-capped mesoporous silica nanoparticles (MSN-DPH). MSN-DPH, preferentially taken up by cancer cells via CD44 receptor-mediated endocytosis, primarily accumulated in mitochondria and efficiently killed cancer cells while exhibiting much lower cytotoxicity to normal cells [143]. In addition, a novel delivery platform based on tetrahedral DNA nanostructures (TDNs) that enable mitochondrial import of Dox for cancer therapy was designed by Yan and colleagues. The peptide 3KLA was conjugated to TDNs to efficiently target mitochondria. The 3KLA-TDNs exhibited highly efficient Dox accumulation in mitochondria, leading to an effective release of cytochrome c and upregulated expression of pro-apoptotic proteins, as well as reduced expression of anti-apoptotic proteins, resulting in activation of mitochondria-mediated apoptotic pathway to enhance the anti-cancer efficacy of Dox [144].

## 6. Conclusions

Mitochondria, with their various functions, have become novel targets for anti-cancer strategies. Targeting mitochondrial metabolism, including the electron transport chain function, the redox signalling pathways and ROS homeostasis, as well as apoptotic signalling pathways, have become a major focus for researchers (Table 1). Mitochondrial DNA was also reported to play a critical role in tumorigenesis; therefore, targeting mtDNA has opened a new direction of anti-cancer therapy. Moreover, delivery of anti-cancer drugs to mitochondria is of high clinical relevance, since it can enhance drug selectivity for cancer cells, overcome drug resistance and considerably promote anti-cancer activity. A prime example is the CI-targeting MitoTam [12] currently under a clinical trial thus far showing excellent therapeutic and toxicity profile. We have finalised Phase 1/b clinical trial of MitoTam and are preparing for Phase 2 trial, likely combining MitoTam with another anti-cancer therapeutic. Overall, the development of mitocans, the mitochondrial-targeting treatments and strategies have great potential in future anti-cancer therapies.

## Figures and Tables

**Figure 1 ijms-21-07941-f001:**
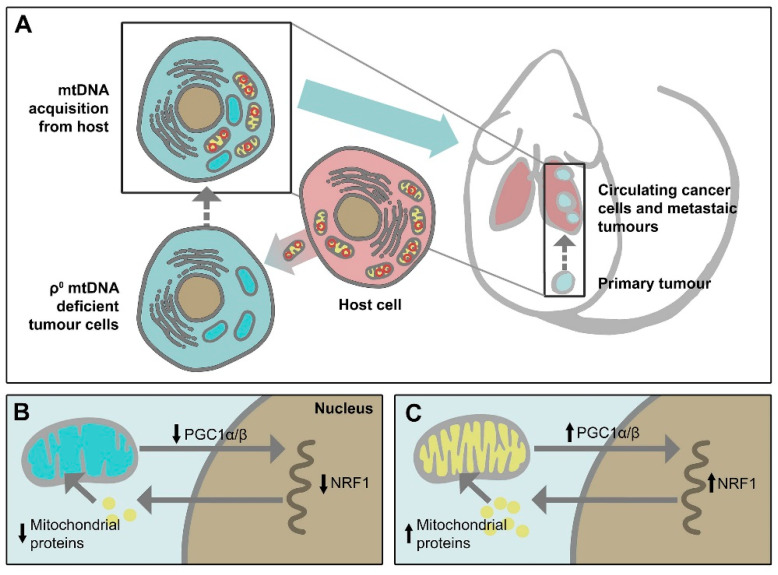
Mitochondrial transfer from host cells leads to tumorigenesis recovery of mtDNA-depleted cancer cells. (**A**) mtDNA deficient ρ0 cancer cells do not form tumours. mtDNA acquisition from host cells leads to recovery of tumorigenic capacity of the cells. (**B**) In mtDNA deficient ρ0 cancer cells, signalling between mitochondria and nucleus is dampened. Reduced levels of the transcription coactivator PGC1α/β leads to the low transcriptional activity of nuclear respiratory factor-1 (NRF1), resulting in the low level of nuclear-encoded proteins imported into the mitochondria and mitochondrial dysfunction. (**C**) Mitochondrial transfer from host cells leads to increased PGC1α/β levels with an increased NRF1 transcriptional activity. This allows appropriate levels of nuclear-encoded mitochondrial proteins to be imported into mitochondria and to recover mitochondrial function.

**Figure 2 ijms-21-07941-f002:**
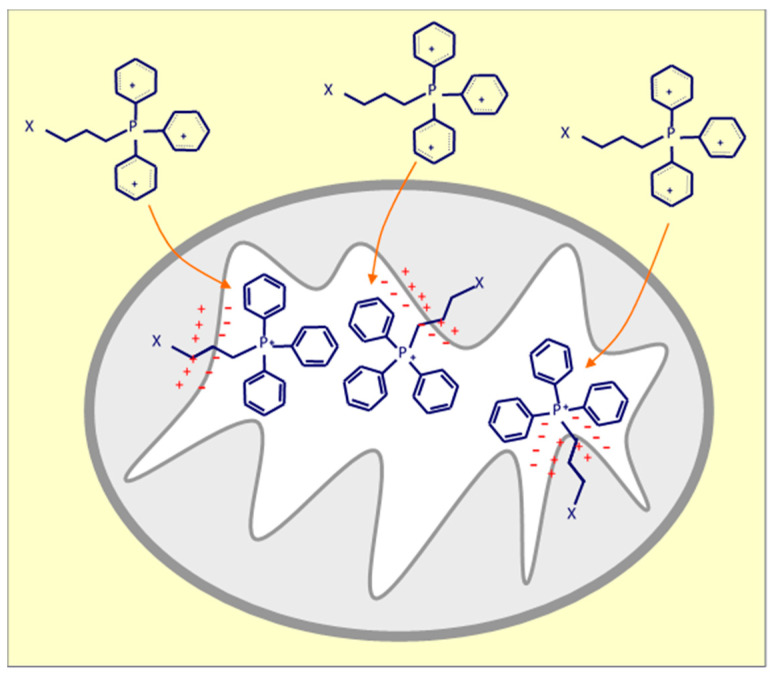
Positively charged triphenylphosphonium (TPP) anchors compound-X in the mitochondrial inner membrane (MIM) due to negative potential at the matrix face of the MIM.

**Figure 3 ijms-21-07941-f003:**
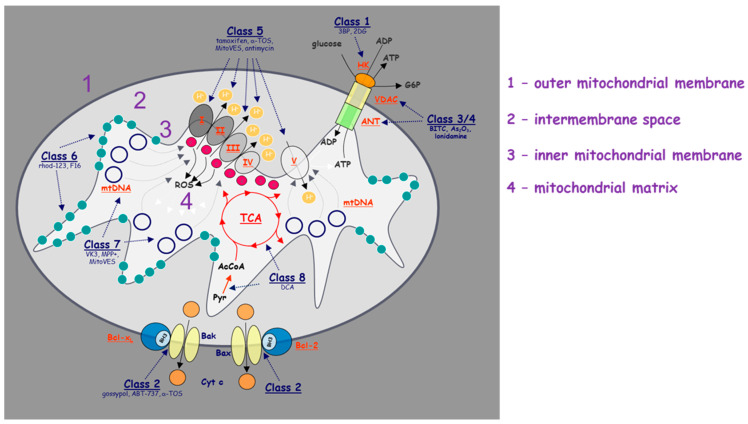
Schematic illustration of the molecular targets of individual classes of mitocans. The classes of mitocans comprise the following, as enumerated from the outside of the mitochondria towards the matrix. Class 1: hexokinase inhibitors; Class 2: BH3 mimetics and related agents that impair the function of the anti-apoptotic Bcl-2 family proteins; Class 3: thiol redox inhibitors; Class 4: agents targeting VDAC and ANT; Class 5: compounds targeting the mitochondrial electron transport chain; Class 6: hydrophobic cations targeting the MIM; Class 7: compounds that affect the TCA; and Class 8: agents that interfere with mtDNA. Class 9 (not shown) includes agents acting on mitochondria, whose molecular target has not been thus far described [10].

**Table 1 ijms-21-07941-t001:** A summary of the anti-cancer drugs discussed in this review and their mechanism of action.

Mitochondrial Function	Drugs	MECHANISM of Action	Types of Tumor	Trial Stage	References
**ETC**	Papuamine	Inhibits ATP production	Lung	NA	[23]
	Metformin	Inhibits Complex I	Colon, lung, ovary, Breast, prostate	Clinical trials	[26,27,28]
	Tamoxifen	Inhibits Complex I	Breast	FDA-Approved	[30]
	MitoTam	Inhibits Complex I	Breast	Clinical trials	[113]
	α-TOS	Inhibits Complex II	Breast	Preclinical	[33,34]
	MitoVES	Inhibits Complex II	Breast	Preclinical	[105]
	VLX600	Inhibits Complex IV	Colon	Preclinical	[36]
	Tigecycline	Inhibits Complex I and IV	Leukemia	FDA-Approved	[37]
	Gamitrinib	Inhibits ATPase activity	Prostate	Preclinical	[38]
**TCA Cycle**	AGI-5198	Inhibits IDHs activity	Glioblastoma	Clinical trials	[12,45]
	Dichloroacetate	Inhibits IDHs activity	Brain	Clinical trials	[12,46]
**Glycolysis and OXPHOS**	2-deoxyglucose (2-DG)	Competitor for binding hexokinase	Lung, prostate, ovary, breast	Clinical trials	[47,49]
	Metformin/2DG	Inhibits ATP production	Lung, pancreas	Clinical trials	[3,47,48]
	ABT737/2DG	Inhibits OXPHOS	Ovary	NA	[49]
**Signalling pathways**	Venetoclax	Bcl-x_L_ inhibitor	Leukemia, lymphoma	FDA-Approved	[16,72]
	Navitoclax	Bcl-Xl/Bcl2 inhibitor	Breast, lung, prostate, colon	Clinical trials	[10,12]
	ECPU-0001	Bcl2 inhibitor	Lung	Preclinical	[73]
	Gossypol	LDHA inhibitor, NADH competitor	Breast, brain, prostate	Clinical trials	[10,28]

ETC = electron transport chain; TCA Cycle = tricarboxylic acid cycle; OXPHOS = oxidative phosphorylation; IDH = isocitrate dehydrogenase; ATP = adenosine triphosphate; LDHA = lactate dehydrogenase; NADH = nicotinamide adenine dinucleotide.

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
