# Peer review of "Mitocans Revisited: Mitochondrial Targeting as Efficient Anti-Cancer Therapy"

_ijms, 2020, doi:10.3390/ijms21217941_

Round 1

Reviewer 1 Report

The authors presented a very interesting review concerning the methods of anti-cancer therapy associated with effects on mitochondrial functions. The review examines in detail the mechanisms of action of a significant number of agents that affect several key signaling and metabolic pathways associated with mitochondria. The authors consider the key pathways of cancer cell metabolism that allow it to actively proliferate, migrate, and change its microenvironment in order to maximize malignancy. Then the main approaches are described to influence these pathways, which can inhibit the growth and invasion of cancer, as well as its resistance to chemotherapeutic effects. The authors provided a lot of information and the review will certainly be of interest to a wide range of researchers.

However, several comments should be addressed to make the presentation more complete and logical.

  1. It is necessary to clearly formulate the definition of "mitocans”. Are any substances that affect the mitochondria, mitocans? Or mitocans are the only substances that have a clearly defined mitochondrial address (for example, TPP-moiety). On the contrary, whether all mitochondrial-targeted compounds are mitocans, or only compounds with anti-cancer activity? All these issues should be clarified convincingly and in detail, so that the classification and identification of such compounds should be not in doubt.
  2. there is a serious contradiction in the authors' description of the role of ROS in cancer cell physiology. The authors state that ROS trigger a number of signal pathways necessary for multiple signaling networks underlying tumor proliferation, survival, and metastatic process. At the same time, many compounds described by the authors as anti-cancer have a key effect on the mitochondria, which causes an increased generation of ROS. In this context, ROS are already being considered as cancer cell killers. Thus, it becomes very unclear whether we need to suppress the pathways of ROS generation in a malignant cell or Vice versa, activate them to prevent its proliferation and metastasis. This is a very important question because the authors in this review do not address the huge body of works devoted to mitochondrial-targeted antioxidants, which many researchers suggested for the treatment of non-cancer pathologies, such as strokes, heart attacks, and others diseases associated with oxidative stress. The question arises whether such mitochondrial antioxidants will be effective in killing cancer cells, or whether they will promote their development.
  3. On the other hand, mitocans, as substances that damage cancer cells, do not have ways to target delivery to the cancer cell, which means they will accumulate in the mitochondria of normal cells as well. This means that it will be possible to damage normal cells, which can cause serious organ damage and severe side effects of such therapy.

All these issues require detailed discussion in the article.

Author Response

We thank the reviewers for their positive comments on our review manuscript. We also appreciate the questions they raised about the term of mitocans, the potential of MitoTAM as well as the role of ROS in both survival signalling and apoptotic killing pathways of cancer cells. Answering and clarifying these questions in the manuscript will definitely further improve the quality of this paper, and provide the following rationale how we have dealt with this.

Reviewer1:

  1. It is necessary to clearly formulate the definition of "mitocans”. Are any substances that affect the mitochondria, mitocans? Or mitocans are the only substances that have a clearly defined mitochondrial address (for example, TPP-moiety). On the contrary, whether all mitochondrial-targeted compounds are mitocans, or only compounds with anti-cancer activity? All these issues should be clarified convincingly and in detail, so that the classification and identification of such compounds should be not in doubt.

‘Mitocan’ is an acronym for ‘mitochondria and cancer’. It first appeared in our review paper published in 2006: Ralph SJ et al (2006) Mitocans: Mitochondria-targeted anti-cancer drugs as improved therapies. Recent Patents Anticancer Drug Discov 1, 327-46. Mitocans are defined as small molecules with anti-cancer activity, whose molecular target is within mitochondria. We classified mitocans into several groups from the surface of mitochondria to their matrix. These groups are: hexokinase inhibitors, compounds targeting Bcl-2 family proteins, thiol redox inhibitors and VDAC/ANT targeting drugs, electron transfer chain targeting drugs, lipophilic cations targeting the inner membrane, drugs targeting the tricarboxylic acid cycle and drugs targeting mtDNA. We published this classification in the following paper: Neuzil J et al, (2013) Classification of mitocans, anti-cancer drugs acting on mitochondria. Mitochondrion 13, 199-208. Since the first appearance of the term mitocans, this acronym has been used in a number of publications. For example, the last publication to date, in which the term mitocans was used, is the following review paper: Mani S, Swargiary G, Singh KK (2020) Natural Agents Targeting Mitochondria in Cancer. Int J Mol Sci 21, E6992. In this paper, the authors (with the senior author Keshav Sing being the editor of Mitochondrion) also bring up the classification of mitocans and highlight the classification of mitocans proposed by us in our 2013 paper mentioned above. The TPP-tagged compounds that we and others have synthetised and tested are in most cases members of class 5 mitocans, i.e. drugs that target the electron transfer chain.

We have added the following statement starting from line 52 in the revised version of the manuscript: ‘We have recently proposed the term ‘mitocans’, an acronym derived from the terms mitochondria and cancer, a group of compounds with anti-cancer activity exerted via their molecular targets within mitochondria, some mitocans being selective for malignant tissues [10]. This classification has been used by others, as for example exemplified by a recent paper [11]. These ...’ Linked to this, we have moved reference number 43 to number 10, added a new reference [11] in the revised version of the manuscript being Mani S, Swargiary G, Singh KK (2020) Natural Agents Targeting Mitochondria in Cancer. Int J Mol Sci 21, E6992. We consequently changed the order of reference number list accordingly. 

  1. There is a serious contradiction in the authors' description of the role of ROS in cancer cell physiology. The authors state that ROS trigger a number of signal pathways necessary for multiple signaling networks underlying tumor proliferation, survival, and metastatic process. At the same time, many compounds described by the authors as anti-cancer have a key effect on the mitochondria, which causes an increased generation of ROS. In this context, ROS are already being considered as cancer cell killers. Thus, it becomes very unclear whether we need to suppress the pathways of ROS generation in a malignant cell or Vice versa, activate them to prevent its proliferation and metastasis. This is a very important question because the authors in this review do not address the huge body of works devoted to mitochondrial-targeted antioxidants, which many researchers suggested for the treatment of non-cancer pathologies, such as strokes, heart attacks, and others diseases associated with oxidative stress. The question arises whether such mitochondrial antioxidants will be effective in killing cancer cells, or whether they will promote their development.

The level of ROS is critical for cell survival and cell death. At moderate concentrations, ROS are mitogenic, also activating pathways linked to cancer cell survival, such as the signaling cascade involving MAPK/ERK1/2, PI3K/Akt, which in turn activate NF-κB, and VEGF. At high concentrations, ROS can cause cancer cell apoptosis (Aggarwal et al., Role of Reactive Oxygen Species in Cancer Progression: Molecular Mechanisms and Recent Advancements. Biomolecules. 2019 Nov; 9(11): 735). Anti-cancer agents, such as MitoTam or MitoVES that can target mitochondrial electron transport chain complex I or II, generate high level of ROS and induce cancer cell apoptosis (Rohlenova et al., Selective Disruption of Respiratory Supercomplexes as a New Strategy to Suppress Her2high Breast Cancer. Antiox Redox Signal. 2017, 26, 84-103; Dong et al., Mitochondrial targeting of vitamin E succinate enhances its pro-apoptotic and anti-cancer activity via mitochondrial complex II. J Biol Chem. 2011, 286, 3717-3728). It is true that there are antioxidants with potential anticancer efficacy, although this field is controversial. As a recent example, we can mention the paper Liu Y et al (2020) Therapeutic targeting of SDHB-mutated pheochromocytoma/paraganglioma (PCPG) with pharmacologic ascorbic acid. Clin Cancer Res 26, 3868-3880. However, publications describing the use of antioxidants as anticancer agents are not really substantiated by clinical practice and in some cases, large-cohort studies indicate that antioxidants are inefficient or, even, deleterious. This is primarily documented by the well-known clinical trial (SELECT), in which the risk of developing prostate cancer was increased by 17% in young men supplemented with vitamin E: Klein EA et al, Vitamin E and the risk of prostate cancer: the Selenium and Vitamin E Cancer Prevention Trial (SELECT). JAMA. 2011 Oct 12;306(14):1549-56. Also, there is a body of literature showing controversial results for the effect of antioxidant polyphenols on cancer cells – some studies show anticancer effect, some show protective effects of these agents. Therefore, we did not focus on antioxidants as anti-cancer agents in this paper, since this would make the review overly long and out of focus.

Notwithstanding the above, we added the following statement in the revised version of the manuscript between line 207-209: ‘The level of ROS is critical for cell survival and cell death. At moderate concentration, ROS activates the cancer cell survival signaling, while high level of ROS can cause damage and induce apoptosis in cells.’

  1. On the other hand, mitocans, as substances that damage cancer cells, do not have ways to target delivery to the cancer cell, which means they will accumulate in the mitochondria of normal cells as well. This means that it will be possible to damage normal cells, which can cause serious organ damage and severe side effects of such therapy.

As mentioned in the answers for question 1, mitocans refers to compounds that target mitochondria and have in some cases selectivity for cancer cells. Therefore, they can selectively kill cancer cells with low toxicity to normal cells/tissues. A prime example is presented by agents that are tagged with the TPP group. These agents, tagged with a lipophilic cationic group, which is a mitochondrial vector, accumulate at the interphase of the inner mitochondrial membrane and mitochondrial matrix. They primarily accumulate in mitochondria of cancer cells, since their mitochondrial inner membrane potential is considerably higher (in negative term) than that of normal cells, being some -180 mV and -60 mV, respectively. These considerations are published in the following paper: Modica-Napolitano JS, Aprille JR (2001) Delocalized lipophilic cations selectively target the mitochondria of carcinoma cells. Adv Drug Deliv Rev 49, 63-70. While the TPP-tagged anti-cancer agents are selective for cancer cells, it is true that not all mitocans, as we defined them, show such level of selectivity.

Reviewer 2 Report

This is a very clear, informative, well-organized and well written review on the possible targeting of tumor cell mitochondria as a possible strategy of specifically eradicating some tumors, notably affecting apoptosis or ROS-related pathways. While the idea mitochondria-based drug targeting (notably using TPP+ or other similar strategies) is not novel, the moving more towards clinical applications is, and the authors do an outstanding job of reviewing the literature and putting it in context, with informative figures and a very useful Table 1. The only thing that is perhaps lacking in the conclusions, is a more detailed and forceful analysis of what the authors consider, in their view, to be the challenges and potential hurdles that are still in the way of a more precise and widespread application of this technology, and what can be done to hopefully solve them. The authors do mention a few issues throughout but given their position and authoritative review of the available data data, I believe this issue could be summarized at the end, and this would be of great interest to readers.

My only other minor concern is that the authors seem to be involved (line 342-44, and the paragraph above) in developing clinical applications of this particular technology, and of a specific compound (Reference 12). This is obviously fine, but this information should be more clearly available (also in the Abstract) as this may create a potential conflict of interest when evaluating the data. The authors declare "no conflict of interest", but, given the above, I beg to differ on this issue.

Author Response

Reviewer 2

This is a very clear, informative, well-organized and well written review on the possible targeting of tumor cell mitochondria as a possible strategy of specifically eradicating some tumors, notably affecting apoptosis or ROS-related pathways. While the idea mitochondria-based drug targeting (notably using TPP+ or other similar strategies) is not novel, the moving more towards clinical applications is, and the authors do an outstanding job of reviewing the literature and putting it in context, with informative figures and a very useful Table 1. The only thing that is perhaps lacking in the conclusions, is a more detailed and forceful analysis of what the authors consider, in their view, to be the challenges and potential hurdles that are still in the way of a more precise and widespread application of this technology, and what can be done to hopefully solve them. The authors do mention a few issues throughout but given their position and authoritative review of the available data data, I believe this issue could be summarized at the end, and this would be of great interest to readers.

We agree with this point that there could be more in terms of challenges and hurdles that prevent broader use of mitocans and related compounds in clinical practice, and that there are attempts to overcome this. However, this is a difficult issue and deserves a separate, in-depth analytical paper to be put together and that would highlight all the various obstacles on the road to translation of mitocans into clinical practice, and we are planning to do so. We added a sentence to the Conclusion as follows (between line 482-485): ‘We have finalised Phase 1/b clinical trial of MitoTam and are preparing for Phase 2 trial, likely combining MitoTam with another anti-cancer therapeutic. Overall, the development of mitocans, the mitochondrial-targeting treatments and strategies have great potential in future anti-cancer therapies.’

My only other minor concern is that the authors seem to be involved (line 342-44, and the paragraph above) in developing clinical applications of this particular technology, and of a specific compound (Reference 12). This is obviously fine, but this information should be more clearly available (also in the Abstract) as this may create a potential conflict of interest when evaluating the data. The authors declare "no conflict of interest", but, given the above, I beg to differ on this issue.

Thanks for the reviewer for pointing out this important point. In fact, we addressed this issue in the Abstract of the manuscript as follows: ‘Mitocans, exemplified by mitochondrially targeted vitamin E succinate and tamoxifen (MitoTam), selectively target cancer cell mitochondria and efficiently kill multiple types of cancer cells by disrupting mitochondrial function, with MitoTam currently undergoing a clinical trial’.

You are correct concerning the ‘conflict of interest’. One of the authors (JN) is involved in the clinical trial and is also a co-owner of the MitoTam intellectual property. This is included in the Conflict of interest statement of the revised version. Thus, rather than saying ‘The authors declare no conflict of interest’, we now state: ‘One of the authors (Jiri Neuzil) is involved in the MitoTam-01 clinical trial (EudraCT 2017-004441-25) and is a co-CEO of MitoTax s.r.o. that is a co-owner of the MitoTam intellectual property.’

Round 2

Reviewer 1 Report

The Authors have made significant improvements to the manuscript following the suggestions of the previous review. The structure of the text, statements, and conclusion have been significantly revised and in the present form, the manuscript sounds good and will be of interest to a fairly wide audience. T